# What Drives the Food Safety Certification: A Qualitative Comparative Analysis of Food Companies in China

**DOI:** 10.3390/ijerph18136893

**Published:** 2021-06-27

**Authors:** Quan Lin, Nana Zhang, Wanchao Guan

**Affiliations:** Department of Business Administration, Business School, Shantou University, Shantou 515063, China; 18nnzhang1@stu.edu.cn (N.Z.); 20wcguan@stu.edu.cn (W.G.)

**Keywords:** food safety certification, institutional theory, qualitative comparative analysis

## Abstract

Food safety is related to health and quality of life. Therefore, the social responsibility of the food industry to ensure food safety has received increasing attention. Influencing factors and mechanisms of participation in food safety certification are important issues in this field. Existing studies on factors affecting food safety certification mainly focus on the impact of a single or few factors. In this study, listed Chinese food companies were selected, and the fuzzy set qualitative comparative analysis method was utilized to explore the matching mode of various factors to encourage companies to obtain food safety certification. The following results were obtained. (1) State-owned companies tend to achieve food safety certification. (2) Companies with high media attention are inclined to achieve food safety certification. (3) In state-owned companies, when the company size is small, a higher degree of food safety concern in the mission statement can encourage the company to achieve food safety certification, and when the company size is large, higher media attention can encourage the company to achieve food safety certification. (4) In non-state-owned companies, when food safety concerns are highlighted in the mission statement and media attention is relatively high, the company will gain food safety certification. This study provides new perspectives on food safety-related research and suggestions for government and the public sector to supervise food safety issues in enterprises.

## 1. Introduction

As the main components of the market at the micro level, companies have played important roles in promoting national economic development and social prosperity. With an increasing number of companies and their increasing social influence, the concept of corporate social responsibility (CSR) has attracted increasing attention worldwide and has been subjected to in-depth research. Food production is the basis of human survival. Food safety is related to the health and livelihood of all members of a society and has a significant impact on the harmonious and stable development of society. Therefore, food companies must focus on food safety issues and accept their social responsibilities. Fuyang inferior milk powder, Sudan red, clenbuterol, and other food safety problems have been repeatedly exposed and have caused significant alarm, and they are considered hidden threats to public safety. China’s government has increased its food industry supervision and, in light of public alarm over food safety incidents, has imposed harsher penalties for unacceptable practices and violations. Recently, the state has gradually improved the laws and regulations related to food safety and achieved certain results in the supervision and management of the food industry. However, food safety problems persist throughout the food industry, and further efforts are required to ensure food safety. Frequent food safety incidents highlight that food companies need to better fulfill their food safety responsibilities and raise questions concerning the factors affecting the fulfillment of such responsibilities.

The concepts of CSR and quality management have common theoretical roots and similar values [1]. Both quality management and CSR have moral and ethical aspects and are extremely similar in terms of effectiveness [2,3]. Food companies are primarily responsible for producing high quality, healthy, and safe food [4]; therefore, certification, such as ISO22000, HACCP, and the ISO9000 series of standards are critical for ensuring food quality and safety.

The reasons a company may aim for food safety certifications differ. Hence, research has been conducted to investigate motivations that encourage certification, which can be divided into internal and external motivations. Internal motivation is from the company management perspective through food safety certification. A company would hope to establish a complete management and production process and form the core competitiveness of the company. External motivation involves companies gaining safety certification as a marketing tool [5] to relieve market pressure or in accordance with relevant laws and regulations. For example, Ruzevicius et al. [6] believed that companies mainly implement quality system certification in accordance with the ISO 9000 series of standards to obtain external advantages.

To investigate the influencing factors of a company’s pursuit of certification, researchers mainly focus on an empirical analysis of a certain system certification and believe that the internal characteristics of the company and external pressure of the market will cause the company to seek ISO 9001 quality certification [7,8,9]. Adams [7] studied the manufacturing industry in New Zealand and believed that when a company is large, the agency problem becomes more serious, and the company will be more likely to obtain ISO 9001 certification. Furthermore, large companies were able to afford certification costs more easily. Terziovski [9] asserted that company performance is related to consumer satisfaction, and the motivation for companies to obtain ISO 9001 certification mainly stems from consumer pressure. The market share of company products and the diversified demands of consumers also affect the implementation of hazard analysis critical control point (HACCP) certification [10]. According to the combinatorial legitimacy theory, Lin et al. [11] proposed that concerns about food safety in company mission statements could encourage companies to achieve food safety certification. In addition, the food safety system, peer pressure, and company management affect the implementation of HACCP certification [12]. These studies mainly focus on the impact of a single or few influencing factors on food safety certification, which is a simple causal logic analysis. This statistical method makes it difficult to clearly describe the interaction between more than three variables. However, the inclination of companies to achieve food safety certification is often affected by the synergistic effects of many factors. The behaviors of food companies need to conform to social norms or cultural cognition in the institutional environment in which the companies are located. Therefore, based on institutional theory and the qualitative comparative analysis (QCA) method, in this study, the matching mode of various factors that encourage companies to achieve food safety certification is explored.

## 2. Theory and Research Model

Institutional theory holds that a company’s environment influences its activities [13]. Therefore, examining the relationship between an organization and its environment, especially its relationship with the institutional environment, can determine and explain the various behaviors of an organization [14]. Margolis and Walsh [15] introduced institutional theory into the field of CSR and proposed that institutional factors are an important motivation for companies to undertake social responsibility. Researchers of institutional theory believe that the institutional environment exerts pressure on companies through certain mechanisms to restrain their behaviors. The behaviors of companies should meet the expectations of society; therefore, the institutional environment requires companies to engage in social responsibility behaviors. Consequently, CSR behaviors cannot be separated from the institutional environment, social culture, etc. [16]. The New Institutional School, which is based on the Three Pillars Theory of Scott [17], holds that institutions include regulatory, normative, and cultural-cognitive elements, as well as related activities and resources that provide a stable meaning for social life. Based on this, Suchman [18] proposed the concept of organizational legitimacy, which indicates that the legitimacy of organizational behavior has been widely recognized from the perspective of social norms or moral values. Most existing studies on CSR behavior from the perspective of institutional theory start from a theoretical perspective on organizational legitimacy. CSR behavior is attributed to the means required to obtain the three dimensions of organizational legitimacy. When an organization’s behavior is consistent with the laws, regulations, social norms, or cultural perceptions in the institutional environment, legitimacy is achieved. When an organization’s behavior is inconsistent with, or significantly deviates from, the expectations of the institutional environment, the organization will face pressure from internal and external legitimacy. In addition, realistic factors, such as the complexity of the institutional environment, scarcity of resources, and limited available resources make it impossible for all companies to simultaneously obtain legitimacy [19,20]. Considering institutional pressure from different sources, companies have different tendencies when obtaining legitimacy. According to the degree of initiative of companies for handling institutional pressure, Oliver [21] classified the legitimacy choices of companies into five types (from positive to negative): compliance, compromise, avoidance, resistance, and manipulation. Similarly, Suchman [18] divided the possible legitimacy preferences of companies into three categories: adapting to the environment, choosing the environment, and controlling and creating the environment. The choice of legitimacy tendency indicates that companies should ensure that their strategic choices for handling external legitimacy can promote the development of companies while needing to obtain the necessary external legitimacy for survival.

Through this theoretical analysis, the fulfillment of the food safety responsibility of companies is the result of the joint action of various internal and external factors. According to internal and external factors of the legitimacy of an organization and conditions of a company, this study finds that the factors that affect a company’s food safety certification are the food safety concerns of the mission statement, media attention, ownership nature, ownership concentration, and company size. Among these factors, media attention and ownership nature, which are external company factors, exert public opinion and policy pressure on companies. A mission statement is the goal and vision set by the company. The degree of ownership concentration reflects the internal ownership structure of a company, and company size represents the operation of a company. These are internal company factors. The five factors used for the analysis are as follows.

### 2.1. Ownership Nature

As an authoritative department, the government is the author of the Food Safety Law and the main department of food safety supervision. Thus, it plays a pivotal role in improving food safety. China’s economic environment enables state-owned companies to master and control the national economy. Therefore, companies with different shareholding characteristics will have various CSR information disclosures. State-owned companies need to fulfill their responsibility to sustain the country’s economic development and lead the industry’s response to government policies [22]. As channels for the government to release information to the outside world, state-owned companies are subject to more pressure from government policies than non-state-owned companies; thus, they are more inclined to fulfill CSR [23]. In terms of social responsibility information disclosure, state-owned companies disclose information more comprehensively, and the quality of disclosure is often higher than that of non-state-owned companies [24]. Therefore, under the condition of more apparent policy and institutional supervision, state-owned companies’ social responsibility performance will be better [25]. For food companies, state-owned companies are more inclined to pursue food safety certification.

### 2.2. Media Attention

Increased media attention on companies will expose managers and the behavior of companies to more public scrutiny. Reputation mechanisms can protect companies from adverse food-safety selection [4]. Considering their corporate reputation, managers and companies attach significant importance to media attention. Any behavior that is harmful to the stakeholders of a company through media reports will damage the company, causing irreparable losses. Perceiving media attention to external legitimacy pressure, companies will choose to comply with the tendency of legitimacy, such as assuming social responsibility [26]. Islam and Deegan [27] investigated the relationship between negative media reports and corporate positive social responsibility information disclosure. They found that when media reports on a certain aspect of social responsibility are more negative, more corporate positive social responsibility information disclosure is required. Their conclusion demonstrates that corporate disclosure of social responsibility information is driven by the legitimacy motive through the disclosure of positive social responsibility information to improve or correct the negative social image created by negative media reports. In this study, the same principle was applied to food safety issues. Under intense media attention, companies are driven by the legitimacy motive and will choose to undertake the behavior of social responsibility, such as attaining food safety certification. Public influence due to media attention on the food safety of companies can influence the behaviors of companies, and it can lead to the government promoting effective legal protection of food safety and enhancing the external legitimacy pressure of companies [28].

### 2.3. Mission Statements Regarding Food Safety Concerns

A mission statement is the basis of strategic decision-making and can help an organization focus on what is important to them [29]. The content of a company’s mission statement reflects its characteristics, priorities, and goals [22]. As the future development direction and cultural symbol of the company, a mission statement is an important source of the cognitive legitimacy of CSR culture. Simultaneously, the mission statement is a signal that can convey the characteristics of a company and a tool for strategic communication among decision makers [30]. It can provide legitimacy to decisions that are in line with the mission and help decision makers share pressure from the board of directors. A mission statement with high food safety concerns is a strong signifier of CSR and clearly conveys the production objectives of the company to all managers and front-line employees, thereby establishing the image of a safe company. Organizational leaders publicly commit themselves to achieving specific goals and ideas and use mission statements to send signals to build the organization’s reputation while publicly disclosing the organization’s commitment to external stakeholders [31]. Although the purpose of emphasizing “security” is to gain external legitimacy and shape the image of social expectation, a clear mission inevitably places normative pressure on organizations and leaders, and the pressure may force companies to make more decisions and actions in favor of food safety.

### 2.4. Ownership Concentration

The ownership of listed companies in China is relatively centralized, and ownership and control rights are often held by one or several major shareholders. The management of a company is often directly appointed by major shareholders, and the values and concepts of management tend to be consistent with those of major shareholders [32]. This mechanism allows large shareholders to influence and control an organization’s management objectives [33]. The legitimacy of a company’s behavior stems from the consistency between the management’s goals and the interests of major shareholders. In the case of frequent food safety incidents, it is difficult for food companies to gain consumers’ trust. Therefore, to improve the image of food companies as good corporate citizens and gain more cognitive legitimacy, major shareholders may be more willing to take more responsibility for food safety [34]. When the shareholding ratio of the largest shareholder is higher, the influence of the largest shareholder on the board of directors and senior management of the company is more significant. Therefore, board members and senior managers can act as trustees for the largest shareholders [35]. Therefore, when the shareholding ratio of the largest shareholder is higher, the influence on important decisions and management behaviors of the company is more significant, and the company will make decisions conducive to food safety.

### 2.5. Company Size

The size of a company is closely related to whether the company undertakes social responsibility because it must focus on and meet the needs of all stakeholders when undertaking social responsibility and fulfilling its obligations. To ensure food safety, food companies must invest a significant amount of human, material, and financial resources to gain relevant safety certification, which occupies a part of the company’s resources. The fulfillment of social responsibility should not affect the normal operation and profitability of a company. However, the size of a company affects whether the company has food safety certification, because companies of different sizes may show various resource allocations, affecting the legitimacy choice of a company [36]. When a company is small, the resources will be more focused on immediate needs, such as cost cutting, and the company may take action to suppress legitimacy, rather than fulfilling more social responsibilities for food safety. According to Trotman and Brandley [37], relatively large companies disclose their social responsibility information in a more complete and professional manner because large companies pay more attention to their reputation. Large-scale companies have more resources, and the cost of fulfilling social responsibilities and gaining safety certification will account for a relatively small proportion of the total cost of the company. Therefore, larger companies are more capable of achieving food safety certification.

This study follows the logical thinking of Campbell et al. [38] and introduces the QCA method (Figure 1).

## 3. Materials and Methods

### 3.1. QCA

QCA research methods first appeared in Charles Ragin’s *Comparative Methods*, which was published in 1987 [39]. This method integrates the advantages of case-based and variable-oriented methods and focuses on the configuration of different combinations of elements. The method considers that a causal asymmetry exists between the condition and result, and multiple paths to produce the same result are possibly available. QCA can effectively compensate for defects in qualitative and quantitative research methods [38]. In addition to being a technical means, it provides a new mode of thinking for academic research [40]. The basis of QCA is set theory and Boolean algebra, and the core logic is set theory. Charles Ragin noted that QCA has the following three characteristics: (1) QCA technology focuses on the multiple concurrent causality of different cases. Multiple implies that the path leading to the outcome is not singular, whereas concurrent indicates that each path is a combination of different antecedent conditions. (2) Moderate universality, that is, QCA analysis, is not limited to simple descriptions but pursues moderate universality. (3) Replicability and transparency (QCA) requires that each case be decomposed into a series of features, comprising a certain number of conditional variables and result variables, with fixed formal rules and replicability [41]. In terms of transparency, choices must be made in a transparent manner in multiple aspects of the analysis, including variable selection, processing, analysis tool selection, and analysis process intervention, and periodically returns to the original case to see its diverse and unique content.

Fiss [42] and other researchers who advocate QCA agree that there may be relationships of complementarity, substitution, and inhibition among the conditional variables of the composition configuration, and the attributes based on the Boolean algebra and set theory of QCA are more suitable for exploring such interactive relations. Each conditional variable that affects the safety certification of food companies has similar or opposite attributes, and complementarity, substitution, and inhibition are possible. The QCA method is more suitable for exploring the relationship between them and the functioning mechanism.

### 3.2. Sample

In this study, and according to the industry classification guidelines of China’s Listed Companies in 2012 [11], the relevant data of China’s food industry listed companies in 2018 were selected as the data support. The sample included food processing and food manufacturing industries. A-share was used as the standard for sample selection, yielding 122 enterprises. Companies with continuous loss, special treatment, or at risk of delisting and missing data were excluded, leaving 115 sample companies for the analysis.

### 3.3. Variable

#### 3.3.1. Outcome Variable

Food safety certification (FQ)

The national certification and accreditation information public service platform provides ISO 9000, ISO 22000, and HACCP certification information. Through this platform, the type, time, and audit information of a company’s certification can be obtained. In this study, the names of companies on the national certification and accreditation information public service were identified, the number of national certifications acquired by a company was queried, and finally, the food safety certification variable was attained. If the company had obtained ISO 22000, ISO 9000, or HACCP certifications, a value of 1 was assigned; otherwise, it was assigned a value of 0.

#### 3.3.2. Condition Variable

(1)Mission statements regarding food safety concerns (MSFC):

In this study, the method of Bartkus and Glassman [30] was applied to investigate the corporate mission statement and to find the text of the corporate mission statement from the annual report and official website of a company. Following David [43] and Klemm et al. [44], this study regarded a company’s statement of mission, purpose, vision, and values as the content of the mission statement, with the company’s business philosophy and company spirit as the supplement. To ensure research accuracy, the mission statement text was cleaned after word segmentation; stop words, such as “of,” “is,” and “we,” were deleted, and the entire mission statement of the company was converted into keyword phrases. The number of occurrences of food safety and synonyms for food safety were then obtained using the dictionary method. The percentage of food safety terms in the company’s mission statement was obtained by comparing the number of occurrences with the number of all keywords mentioned in its mission statement to measure the mention of food safety in the mission statement.

(2)Media attention (Medatt):

Media attention data were obtained by manually collecting and collating the reports of companies from the CNKI China Major Newspapers Full-Text Database [45]. According to previous studies, the number of newspaper reports on companies is often collected as a metric of media attention [46,47], and media attention was defined as the number of times the company was reported by the media. The names of listed companies were entered into the database, and a value of 1 was added to the number of news reports of listed companies. The natural logarithm, that is, the measurement index of media attention in this study, is represented by Ln (total number of media reports + 1).

(3)Ownership nature (OWN):

Property-right nature is a dichotomous variable. The data of listed companies disclosed by the CSMAR database contain information about the ultimate controller of the company, which can determine whether the company is a state-owned holding company. According to the definition of the ultimate controller type of the listed company in the CSMAR database, if the ultimate controller of the company was state-owned, it was defined as a state-owned company, and the value of OWN was 1. If the ultimate controller of the company was not state-owned, it was defined as a non-state-owned company, and the value of OWN was 0.

(4)Ownership concentration (First):

Ownership concentration reflects the degree of influence of major shareholders on corporate decision-making. Following Wu [48], the documents of the top 10 shareholders were downloaded from the national CSMAR database, and the proportion of common shares held by the largest shareholder at the end of the year was selected as the measurement index of ownership concentration. This measure was used to measure the degree of control for the largest shareholder.

(5)Company size (Size):

Following Wang [45], the company asset load table from the CSMAR database was downloaded, and the total assets at the end of the year were selected. The company scale was measured by the natural logarithm of the total assets, that is, the company size = ln (total assets at the end of the year).

The definitions and data sources for all variables are shown in Table 1.

## 4. Results

### 4.1. Descriptive Statistics

Before the QCA, a descriptive statistics analysis was conducted on the original data using SPSS24.0 (Table 2). The average value of the food safety certification data was 0.7478. This indicates that nearly 74.8% of food processing companies’ products have obtained food safety certification, and 86 companies (115 companies in the sample) obtained food safety certification. In the mission statement, the maximum value of food safety concern was 1, the mean value was 0.0728, and the standard deviation was 0.1055. The mean value of media attention was 1.8234, the maximum value was 5.7991, and the minimum value was 0. The average value of the property right nature data was 0.3739, indicating that nearly 33.4% of the sample companies belong to state-owned companies. The mean value of the ownership concentration data was 0.3712, the maximum value was 0.7966, and the minimum value was 0.1041, indicating that the control intensity of major shareholders varied significantly among companies. The mean value of the company size data was 22.2461, the maximum value was 26.7975, the minimum value was 20.3607, and the mean value was 1.0701, indicating that a significant difference exists in the size of the listed food companies in the sample.

### 4.2. Calibration of Outcome Variables and Conditional Variables

A QCA can be divided into a clear set QCA, a multi-valued set QCA, and a fuzzy set QCA (fsQCA). A clear QCA requires the conversion of all conditions and outcome data to 0 or 1. The fsQCA (University of California, Irvine, CA, USA) requires converting the data values of the outcome and conditions into corresponding membership degrees, which are between 0 and 1. When the membership value of the fsQCA is closer to 1, the membership value will be higher, and vice versa. The calibration method used the COMPUTE program under the variables directory on fsQCA software, calibrated the data using the Calibrate function, and converted the data to a membership score. This process needed to set three thresholds for each condition and result in the Calibrate program (x, n1, n2, and n3). Here, x is the condition or result, n1 is the threshold of full membership, n2 is the threshold of the crossing point, and n3 is the threshold of complete non-membership. The specific setting of the threshold value should be based on the corresponding theory. In the absence of explicit theoretical guidance, mechanical cutoff points can be used according to the characteristics of the data.

In this study, variable calibration was based on literature or theory. In the absence of clear literature or theoretical references, a mechanical cutoff point was used. The outcome variable of food safety certification and the condition variable of ownership nature were dichotomous variables (where 1 denotes food safety certification, 0 denotes non-food safety certification, 1 denotes state-owned companies, and 0 denotes non-state-owned companies). The mission statement of food safety awareness, media attention, ownership concentration, and company size were combined with measurement methods and descriptive statistics, using the method of mechanical anchor point calibration. The 95% quantile value of the data was used as the threshold for full membership, the 5% quantile value as the threshold for full non-membership, and the 50% quantile value as the intersection. The specific calibration thresholds of the results and the conditional variables are listed in Table 3.

### 4.3. Analysis of the Necessity of Single Causal Conditions

According to the mainstream QCA research process, this study first determined whether a single antecedent condition (including non-set) was a necessary condition for companies to obtain food safety certification before conducting the configuration analysis. In QCA studies, if there is a condition variable in all the configurations that make the outcome variable occur, this condition is considered necessary for the result to occur [49]. Necessary condition identification was measured based on the consistency level of all conditions. Only when the consistency level of a condition variable is greater than 0.9, can it be considered a necessary condition for the occurrence of a result [40,41,49]. The fsQCA3.0 software (University of California, Irvine, CA, USA) package was used for data processing, and the results are shown in Table 4. The consistency level of the five antecedent conditions did not exceed 0.9. This indicates that there is no necessary condition for companies to obtain food safety certification among the five condition variables (Table 4).

### 4.4. Truth Table Analysis

In contrast to a regression analysis, the focus of QCA is not the correlation between conditions and results, but the adequacy of results caused by different configurations of multiple conditional variables. Previous studies agree that the configuration is accepted only when the consistency level of adequacy is greater than or equal to 0.75 [40]. According to different research problems and scenarios, the selection of the consistency threshold can also have different values, such as 0.76 [41] and 0.8 [50]. The frequency threshold is related to the sample size. Generally, the frequency threshold of small and medium samples is set to 1, and that of large samples is set to 2 [39]. In specific research, the distribution of cases in the truth table and researchers’ familiarity with the sample cases need to be considered. In this study, configuration balance and reduced potential contradiction configurations were considered. The minimum consistency with PRI should be ≥ 0.75 (0.7, which is also acceptable) [50]. The consistency threshold used in this study was 0.8, and the frequency threshold was 1.

After the consistency threshold and frequency threshold were set, the configuration combination analysis results were obtained. The fsQCA3.0 program delivers three types of solutions with different complexities: complex, reduced, and intermediate. Based on previous studies, this study reports simple and intermediate solutions [42]. Table 5 shows the result of the simple solution output by the fsQCA program, and the condition variables in the simple solution are the core conditions. Table 6 shows the result of the intermediate solution output by the fsQCA program, which shows the specific configuration and related data. Table 7 shows the configuration formed by the five condition variables of the intermediate solution and its results for food safety certification.

FsQCA was adopted in this study. The detailed analysis steps are shown in Figure 2.

## 5. Discussion

### 5.1. Analysis of Combinations

The fsQCA program effectively identified four configurations for Chinese food companies to obtain safety certifications. This implies that the combination and matching mode of the company’s safety certification is not singular but is diversified.

First configuration in Table 7, solution 1 (MSFC*OWN*~First*~Size): In this configuration, “MSFC” and “Ownership nature” exist, while “Ownership concentration” and “Company size” are absent. This configuration has the highest consistency level (0.89) and raw coverage rate (0.04), covering five cases. This configuration shows that, in state-owned companies, the mission statement of companies focus more on food safety; when the company is small and the ownership concentration is relatively low, the company will choose to obtain food safety certification.

Second configuration in Table 7, solution 2 (MSFC*Medatt*~OWN*First*~Size): In this configuration, “MSFC,” “Media attention,” and “Ownership concentration” exist, while “Ownership nature” and “Company size” are absent. The consistency level of this configuration was 0.83, the raw coverage rate was 0.01, and six cases were covered. In this configuration, the companies that had obtained safety certification were non-state-owned companies, and the media attention and ownership concentrations were relatively high; however, the scale of the company was relatively small. This configuration suggests that, if a non-state-owned company pays great attention to food safety in the disclosed mission statement, the media pay high attention, and if the equity is relatively concentrated, the company is inclined to obtain food safety certification.

Third configuration in Table 7, solution 3a (Medatt*OWN*First*Size): In this configuration, “Media attention,” “Ownership nature,” and “Company size” exist as the core conditions, and “Ownership concentration” exists as the auxiliary condition. The consistency level of this configuration was 0.84, the raw coverage rate was 0.04, and 11 cases were covered. This configuration shows that, in state-owned companies, higher media attention, greater equity concentration, and larger company size can prompt companies to achieve food safety certification.

Fourth configuration in Table 7, solution 3b (~MSFC*Medatt*OWN* Size)). In this configuration, “Media attention,” “Ownership nature,” and “Company size” exist as the core conditions, and “MSFC” as the auxiliary condition is absent. The consistency level of this configuration was 0.81, the raw coverage rate was 0.01, and nine cases were covered. The difference between solutions 3a and 3b is that equity is relatively dispersed, and the mission statement is less concerned about food safety. This shows that, in state-owned large-scale companies with high media attention, when the focus on food safety in the mission statement of the company is relatively low, the company tends to obtain food safety certification.

### 5.2. Result Analysis

#### 5.2.1. Role of Ownership Nature and Media Attention on Food Safety Certification

The government is the authoritative organization of the country, and the media addresses a larger audience in society. Researchers have noted that these two organizations play a pivotal role in the supervision of companies while fulfilling their social responsibilities and ensuring safety. When examining a single condition (horizontal) in Table 7, it was found that media attention and ownership nature appeared three times in the four configurations. This indicates the importance of these two conditions for a company’s food safety certification. The government is the promulgator of food safety laws and regulations, as well as the main regulatory authority for food safety. The governance and management structure of state-owned companies reflects the government’s will and is subject to the greatest degree of institutional pressure from the government. Unlike non-state-owned companies, state-owned companies will cooperate with the implementation of government policies to a greater extent and adopt more social responsibilities to express the government’s attitude toward society. Therefore, state-owned companies tend to obtain external legalities for food safety certification. Media attention keeps the company under scrutiny by society and the public. In particular, positive media reports play a positive role in propaganda for the company, whereas negative media reports will cause the company to lose its credibility and cause major crises. Therefore, the media will place greater external legal pressure on companies in terms of public opinion. Media attention works through the reputation mechanism, making it difficult for companies to establish and maintain their “legitimate” image. Therefore, companies will be more inclined to attain relevant safety certifications.

#### 5.2.2. Dual Role of Company Size in Food Safety Certification

Whether for CSR or other purposes, the company’s decisions must not affect its normal operations. In contrast, a company’s decisions should be beneficial to its operations. Companies need to invest a significant amount of human, material, and financial resources to obtain food safety certifications. The results of this study show that companies at relatively large or small scales are likely to engage in food safety certification behaviors; however, the reasons for specific behaviors may differ.

In the authentication configuration of larger companies (configurations 3a and 3b), the media attention of companies is relatively high, and they are all state-owned companies. Large-scale companies have a relatively high market share of their products and a wide range of company influences, which will attract more attention from the news media. Further, greater media attention and exposure to the company’s products and activities exert external pressure on the company. Therefore, corporate managers tend to fulfill their social responsibilities and ensure food safety in light of media pressure. State-owned companies need to fulfill their responsibility to sustain the country’s economic development and to lead the industry to respond to government policies. Larger state-owned companies have greater responsibility in this regard and are subject to a higher degree of government control. Therefore, large-scale state-owned companies are more likely to follow the requirements of policies, laws, and regulations to fulfill their social responsibilities. In summary, large-scale companies are mainly affected by external pressures, such as the government and media, for food safety certification.

In the authentication configuration of smaller companies (configurations 1 and 2), the MSFC is relatively high. A mission statement is the voluntary text of the company that can reflect the business vision and purpose of the company and belongs to the internal factors that influence the company to obtain food safety certification. High concern about food safety in the mission statement indicates that the company managers attach significance to food safety issues and have a high willingness to fulfill social responsibilities. The company in configuration 1 is a state-owned company. Although the company in configuration 2 is a non-state-owned company, it has received relatively high media attention. Ownership and media attention are external factors that affect the food safety certification of companies. The relatively high attention paid to state-owned companies and media indicates that companies are under high external pressure. Therefore, the food safety certification behavior of relatively small companies is driven by both internal and external factors.

#### 5.2.3. Role of External Factors and Internal Situational Factors in Different Property Rights

From the perspective of the relationship between the configurations (vertical–horizontal two-way) in Table 7 in configurations 1, 3a, and 3b, the companies that obtain food safety certification are all state-owned. Among the relatively small state-owned companies, those with higher MSFC are inclined to obtain food safety certification. Among the state-owned companies with low MSFC, those that are larger in scale and have high media attention are inclined to obtain food safety certification. The analysis of internal and external factors revealed that media attention is an external regulatory factor, and the mission statement food safety attention as the content of the company’s disclosure document is an internal factor affecting the company.

A high degree of food safety concern in the mission statement indicates that the management and strategic decision-making of the company focuses more on issues of food safety and consumer health. To maintain their reputation, companies will constrain their behavior and take more action to comply with social norms and gain legitimacy, thus taking measures to ensure food safety. Therefore, the company’s strong internal motivation will prompt the company to take the initiative to achieve food safety certification. Although the mission statement in the information disclosed by the company has a low degree of food safety concern, it does not imply that the company pays no attention to ensuring food safety. However, it does at least show that the measures to ensure food safety are not the focus of the company’s strategic decisions. When the internal motivation of the company is relatively low, higher media attention exposes the company to the public. Negative media reports will damage a company’s reputation and potentially cause a crisis. Conversely, positive media reports result in good publicity, which can improve a company’s reputation and increase consumer trust. When facing high external concerns, companies will choose to obtain food safety certifications that comply with external legality. With the help of media publicity, the company’s products are more trusted, and a positive image is established. This shows that, in state-owned companies, media attention as an external factor and mission statement food safety attention as an internal situational factor have a relatively subtle effect. External factors or internal situational factors can encourage state-owned companies to obtain food safety certifications.

Configuration 2 is a combination of conditions for non-state-owned companies to obtain food safety certifications. A company is a profitable organization, and its goal is to maximize shareholder equity. Food safety certification is a complex process that requires many resources in the early stages of certification. It also needs to be audited every year after passing the certification; consequently, the reasons affecting the food safety certification behavior of non-state-owned companies will be more complex. According to the configuration, non-state-owned companies require a mission statement for safety certification, food safety attention, and media attention. It is only when the company has the internal motivation to undertake food safety, and when external supervision is relatively strict, the company can make safety investments. Therefore, when comparing the substitution effect of external influencing factors and internal situational factors in state-owned companies, non-state-owned companies will achieve food safety certification only when internal and external factors interact.

## 6. Conclusions

In the context of increasingly serious food safety problems in China, it is crucial to actively explore methods for improving food safety and to consider the mechanism of multiple factors that affect the safety certification of food companies. From the perspective of set theory, this study obtained different configurations of food safety certification for companies through an fsQCA. The results of this study reveal that most configurations reflect the importance of ownership nature and media attention. The specific conclusions are as follows: (1) Compared with non-state-owned companies, state-owned companies are more inclined to obtain food safety certification, and the combination of antecedents and conditions for state-owned companies to gain food safety certification is more diversified; (2) companies with higher media attention are also more inclined to obtain food safety certification; (3) the effect mechanism of the company scale on food safety certification is different; (4) in state-owned companies, a higher MSFC can promote food safety certification of companies in the context of small-scale companies; in the case of large-scale companies, higher media attention can encourage companies to achieve food safety certification; (5) in non-state-owned companies, the combination of the antecedent factors that both MSFC and media concerns exist can promote food safety certification.

Our results have several policy implications. The external supervision of companies is an important way to ensure food safety. This finding indicates that the external pressure perceived by companies is of great significance to improving food safety. Therefore, we can strengthen the implementation of policies, increase the binding force of policies on companies, and impose greater policy pressure on companies. Simultaneously, the media’s attention has a very good supervision effect for companies: companies hope to establish a good reputation through the media, with whom the government can cooperate to exert pressure on companies through law and public opinion to promote food safety certification. In addition, it can be seen that a company’s mission statement has a constraining effect on managers. It is worth noting that among small companies, both state-owned and non-state-owned, their mission statements pay more attention to food safety, indicating that small companies have a strong intrinsic motivation to adopt social responsibility. However, large-scale companies mainly rely on external supervision, which is a matter worth considering. As large state-owned food companies play an important role in social stability, governments must strengthen the education of the managers of large state-owned companies and improve their sense of social responsibility and dedication to ensure food safety. In addition, the government should introduce policies to encourage small- and medium-sized entrepreneurs to carry out food safety certification or other reform measures, and reward companies with good performance in tax, capital or other aspects, thereby promoting companies to set up a model of “good companies “ and encourage them to innovate in environmental protection.

This study has some limitations. First, owing to the limitations of the QCA methods and data sources, the conditional variables selected in this study were limited, and the overall number of research samples was relatively small. Future research can expand the conditional variables, increase the number of samples, and further explore the mechanisms that affect the safety certification of food companies. Second, the sample selected in this study is sourced from China’s listed food companies, which use sufficient economic strength and motivation to formulate mission statements and issue annual reports, CSR reports, sustainable development reports, and food safety certifications. Owing to the large number of food companies in China, the reasons for food safety certification of unlisted small- and medium-sized food companies may be more diversified. The conclusion of this study is not general, and it is difficult to apply it to unlisted small- and medium-sized companies. Future studies should use non-listed food companies as samples and adopt additional methods, such as questionnaire surveys, to further study the motivation of companies to engage in food safety responsibility.

## Figures and Tables

**Figure 1 ijerph-18-06893-f001:**
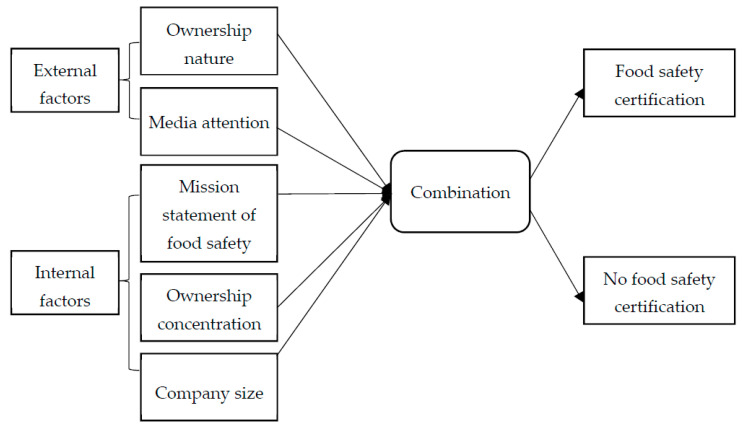
Research model.

**Figure 2 ijerph-18-06893-f002:**
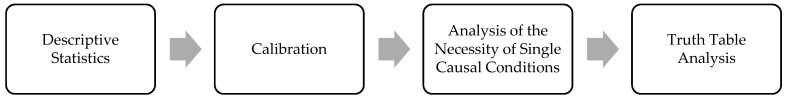
Flow chart of data analysis.

**Table 1 ijerph-18-06893-t001:** Description and source of outcome and conditional variables.

	Variables	Description	Data Sources
Outcome	FQ	If there are ISO 22000, ISO 9000, or HACCP certifications, the value is 1; otherwise, it is 0	National certification and accreditation information public service platform
Condition	MSFC	The proportion of food safety in the total field of mission statement	Company annual report, official website
	Medatt	Ln (total number of media reports + 1)	CNKI full text database of important Chinese newspapers
	OWN	For state-owned companies, the value is 1; for non-state-owned companies, the value is 0	CSMAR
	First	Shareholding ratio of the largest shareholder	CSMAR
	Size	ln (total assets at the end of the year)	CSMAR

**Table 2 ijerph-18-06893-t002:** Descriptive statistics.

Variables	N	Min	Max	Mean	STD
FQ	115	0.0000	1.0000	0.7478	0.4362
MSFC	115	0.0000	1.0000	0.0728	0.1055
Medatt	115	0.0000	5.7991	1.8234	1.3276
OWN	115	0.0000	1.0000	0.3739	0.4860
First	115	0.1041	0.7966	0.3712	0.1399
Size	115	20.3607	25.7975	22.2461	1.0701

**Table 3 ijerph-18-06893-t003:** Calibration thresholds of outcome and conditions.

Variables	Calibration		
	Complete Membership Point	Intersection Point	No Membership Points At All
FQ	1.0000	/	0.0000
MSFC	0.2000	0.0556	0.0000
Medatt	4.2073	1.7918	0.0000
OWN	1.0000	/	0.0000
First	0.5944	0.3579	0.1683
Size	24.2883	22.0639	20.8674

**Table 4 ijerph-18-06893-t004:** Necessity analysis of single antecedent conditions.

Condition Variables	Obtain Food Safety Certification		Condition Variables	Obtain Food Safety Certification	
	Consistency	Coverage		Consistency	Coverage
MSFC	0.4736	0.7874	~MSFC	0.5264	0.7155
Medatt	0.4986	0.7877	~Medatt	0.5014	0.7120
OWN	0.3721	0.7442	~OWN	0.6279	0.7500
First	0.5123	0.7645	~First	0.4877	0.7310
Size	0.4877	0.7613	~Size	0.5123	0.7354

Note: The condition variable in the left column indicates the existence of the condition, and the condition variable in the right column indicates the absence of the condition.

**Table 5 ijerph-18-06893-t005:** Simple solution of fsQCA output (consistency: 0.8).

--- Parsimonious Solution ---			
frequency cutoff: 1			
consistency cutoff: 0.814711			
	raw coverage	unique coverage	consistency
Medatt*OWN*Size	0.17407	0.125698	0.839126
MSFC*OWN*~First*~Size	0.0848837	0.0365117	0.877404
MSFC*Medatt*~OWN*First*~Size	0.111047	0.111046	0.83479
solution coverage: 0.321628			
solution consistency: 0.845355			

**Table 6 ijerph-18-06893-t006:** Intermediate solution of fsQCA output (consistency: 0.8).

--- Intermediate Solution ---			
frequency cutoff: 1			
consistency cutoff: 0.814711			
	raw coverage	unique coverage	consistency
MSFC*OWN*~First*~Size	0.0848837	0.0380233	0.877404
~MSFC*Medatt*OWN*Size	0.116628	0.0122094	0.814785
Medatt*OWN*First*Size	0.145698	0.037093	0.83925
MSFC*Medatt*~OWN*First*~Size	0.111047	0.111046	0.83479
solution coverage: 0.309651			
solution consistency: 0.845129			

**Table 7 ijerph-18-06893-t007:** Configuration for obtaining food safety certification (consistency: 0.8).

Conditional Variable	Solution			
	1	2	3a	3b
MSFC	●	●		⊗
Medatt		●	●	●
OWN	●	⊗	●	●
First	⊗	●	●	
Size	⊗	⊗	●	●
Consistency	0.877404	0.83479	0.83925	0.814785
Raw coverage	0.084884	0.111047	0.145698	0.116628
Unique coverage	0.038023	0.111046	0.037093	0.012209
Solution coverage	0.309651			
Solution consistency	0.845129			

Note: ● = the core condition exists, ⊗ = the core condition is absent, ● = the auxiliary condition exists, and ⊗ = the auxiliary condition is absent. A blank space indicates that the condition can exist or be absent.

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
