# Peer review of "What Drives the Food Safety Certification: A Qualitative Comparative Analysis of Food Companies in China"

_ijerph, 2021, doi:10.3390/ijerph18136893_

Round 1

Reviewer 1 Report

[Abstract] It is missing here in the " Abstract" the indication of the methodology followed by the authors, and also some of the values obtained at the end of the investigation.

[line 56] The authors refer to the "implementation of ISO 9001 by companies". But organizations cannot implement ISO 9000 as it is a vocabulary standard for quality management... The certification standard is the ISO 9001 standard! It would be important for the authors to correct this situation.

[line 56] The authors refer to "certification by the QS standard". Which standard is QS? Are the authors referring to the QS 9000 standard (quality standard for the automotive sector???????) It is important that the authors correctly indicate which normative reference they are referring to, so that there are no doubts whatsoever.

[line 71-73] It would be important at this point in the article to also address ISO 22000, as it makes no sense to appear only in the Case Study... As this research is about certification in the food industry, why didn't the authors also address certification by the ISO 22000 standard in this research? It would have brought more interest to researchers in the food industry...

[line 152-154] The authors make the following statement: “Perceiving media attention to external legitimacy pressure, companies will choose to comply with the tendency of legitimacy, such as assuming social responsibility.” - This statement is very strong and important, but what did the authors base this information on? Are these the authors' perceptions or were they based on facts (publications)???

The testing/analysis of the results obtained is very confusing and should be improved. Why didn't the authors make some graphs for the readers to better understand the data analysis and ediuctions?

[line 257-263] It would be important that at this point in the article, that the authors indicate the sample size used in the study, and not just indicate how many companies they excluded...

[line 553-555] The authors make the following statement: “In the context of increasingly serious food safety problems in China, it is crucial to actively explore methods for improving food safety and to consider the mechanism of multiple factors that affect the safety certification of food companies.”. But what does this article contribute to solving this problem? Is the article only an analysis of the current situation? More was expected... It is recommended to the authors to improve this situation.

What are the authors' recommendations for future research?

Author Response

Comments and Suggestions for Authors

[Abstract] It is missing here in the " Abstract" the indication of the methodology followed by the authors, and also some of the values obtained at the end of the investigation.

Thank you for your suggestion. The research value of this study has been added at the end of the abstract on page 1, lines 20-22.

[line 56] The authors refer to the "implementation of ISO 9001 by companies". But organizations cannot implement ISO 9000 as it is a vocabulary standard for quality management... The certification standard is the ISO 9001 standard! It would be important for the authors to correct this situation.

Thank you for your suggestion. ISO9000 is a quality management system with series of certifications. We have made the corresponding modifications on page 2, lines 58-59.

[line 56] The authors refer to "certification by the QS standard". Which standard is QS? Are the authors referring to the QS 9000 standard (quality standard for the automotive sector???????) It is important that the authors correctly indicate which normative reference they are referring to, so that there are no doubts whatsoever.

Thank you for your suggestion. We have corrected this on page 2, lines 58-59.

[line 71-73] It would be important at this point in the article to also address ISO 22000, as it makes no sense to appear only in the Case Study... As this research is about certification in the food industry, why didn't the authors also address certification by the ISO 22000 standard in this research? It would have brought more interest to researchers in the food industry...

Thank you for your suggestion. The ISO22000 certification, which is the same as other certifications, was our measure of the result variables. As this article does not introduce the meaning of certification, therefore, it is not mentioned. We have added this to our manuscript on page 2, lines 48-49.

[line 152-154] The authors make the following statement: “Perceiving media attention to external legitimacy pressure, companies will choose to comply with the tendency of legitimacy, such as assuming social responsibility.” - This statement is very strong and important, but what did the authors base this information on? Are these the authors' perceptions or were they based on facts (publications)???

Thank you for your suggestion. We have added a reference on page 4, line 160.

The testing/analysis of the results obtained is very confusing and should be improved. Why didn't the authors make some graphs for the readers to better understand the data analysis and ediuctions?

Thank you for your suggestion. We have added a data analysis flowchart presented as Figure 2 on page 11. The conclusions drawn in Chapter 5 are based on an analysis of the data represented in Table 7.

[line 257-263] It would be important that at this point in the article, that the authors indicate the sample size used in the study, and not just indicate how many companies they excluded...

Thank you for your suggestion. We made the corresponding changes on page 6, lines 272-273.

[line 553-555] The authors make the following statement: “In the context of increasingly serious food safety problems in China, it is crucial to actively explore methods for improving food safety and to consider the mechanism of multiple factors that affect the safety certification of food companies.”. But what does this article contribute to solving this problem? Is the article only an analysis of the current situation? More was expected... It is recommended to the authors to improve this situation.

Thank you for your suggestion. We have added some policy implications on page 14, lines 583-603.

What are the authors' recommendations for future research?

Thank you for your suggestion. Suggestions for future research can be found on page 15 (lines 615-617).

Reviewer 2 Report

The manuscript ‘What Drives the Food Safety Certification: A Qualitative Comparative Analysis of Food Companies in China’ is the study to identify factors that drives the decision to participate in food safety certification mechanisms in China. The subject of the report is important from consumer point of view – one has to be convinced that food consumed every day is safe and doesn’t cause any unexpected results. The study needs few improvements before publication in IJERPH journal.

COMMENTS:

  • Please add the diagram to visualize the entire process of analysis describe in Results section. It is difficult for a reader to follow pure description.
  • Authors based on 115 samples. Please show the efficiency of the classification model constructed based on the data together with features as specificity, sensitivity, confusion matrix, etc.
  • Please show results of the analysis of training set regarding attributes used (correlation, minimization strategy, etc.)
  • 115 samples for Chine represent only small amount of the possible instances and probably the set is not representative for Chine. Please justify your statement (115 samples are enouh) or identify additional samples and rebuild and test your classification model.
  • It seems that there are three main factors behind food certification: company revenues, reputation and political issues (the government own companies are responsible to assure that the food market is filled in minimal range). Improve your discussion.
  • Please list your finding in Conclusion section in a simple and understandable way.

Author Response

Comments and Suggestions for Authors

The manuscript ‘What Drives the Food Safety Certification: A Qualitative Comparative Analysis of Food Companies in China’ is the study to identify factors that drives the decision to participate in food safety certification mechanisms in China. The subject of the report is important from consumer point of view – one has to be convinced that food consumed every day is safe and doesn’t cause any unexpected results. The study needs few improvements before publication in IJERPH journal.

COMMENTS:

Please add the diagram to visunawoalize the entire process of analysis describe in Results section. It is difficult for a reader to follow pure description.

Thank you for your suggestion. We have added a data analysis flowchart as Figure 2 on page 11. The conclusions drawn in Chapter 5 are based on an analysis of the data represented in Table 7.

Authors based on 115 samples. Please show the efficiency of the classification model constructed based on the data together with features as specificity, sensitivity, confusion matrix, etc.

Thank you for your suggestion. This study hopes to determine the combination of antecedent conditions needed to promote food safety certification in food enterprises using a qualitative comparative analysis method. FSQCA software can directly obtain the combination of various factors. Since there is no prediction of combination conditions, there is no confusion matrix analysis of data. Simultaneously, according to the previous studies using the QCA method, a confusion matrix is not required in this paper.

Please show results of the analysis of training set regarding attributes used (correlation, minimization strategy, etc.)

Thank you for your suggestion. The correlation of each variable is shown in the table below. Referring to previous studies that used the QCA method, we found that correlation and minimization strategies were not the main focus of our study. As such, these have not been reported in this paper.

Variable

OWN

MSFC

Medatt

First

Size

OWN

1

MSFC

-0.081

1

Medatt

0.135

0.148

1

First

0.220**

0.023

0.026

1

Size

0.162*

-0.001

0.490***

0.125

1

Note: ***, **, * denote statistical significance at 1%, 5% and 10%.

115 samples for Chine represent only small amount of the possible instances and probably the set is not representative for Chine. Please justify your statement (115 samples are enouh) or identify additional samples and rebuild and test your classification model.

Thank you for your suggestion. Our research is based on listed food enterprises in China. The sample includes all listed food enterprises in China after excluding enterprises with missing data and enterprises that have been delisted; therefore, this study is representative of food enterprises in China.

It seems that there are three main factors behind food certification: company revenues, reputation and political issues (the government own companies are responsible to assure that the food market is filled in minimal range). Improve your discussion.

Thank you for your suggestion. These factors have been fully discussed in section 5.2 (Result Analysis) in the manuscript.

Please list your finding in Conclusion section in a simple and understandable way.

Thank you for your suggestion. We have revised the corresponding text in the revised manuscript on page 14, lines 571-581.

Round 2

Reviewer 2 Report

Dear Authors,

The manuscript can be published in present form.